# Behavior Change in Chronic Health: Reviewing What We Know, What Is Happening, and What Is Next for Hearing Loss

**DOI:** 10.3390/ijerph20085605

**Published:** 2023-04-21

**Authors:** Sophie Brice, Helen Almond

**Affiliations:** 1Department of Nursing and Allied Health Sciences, Faculty of Health Arts and Design, Swinburne University of Technology, Hawthorn, VIC 3122, Australia; 2The Australian Institute of Health Service Management, College of Business and Economics, The University of Tasmania, Hobart, TAS 7000, Australia

**Keywords:** adherence, adoption, behavior change, complex chronic health, consumer-centered digital health, hearing care, age-related sensorineural hearing loss, intervention, person-centered

## Abstract

Untreated age-related sensorineural hearing loss is challenged by low adoption and adherence to hearing aids for treatment. Hearing care has evolved from traditional clinic-controlled treatment to online consumer-centered hearing care, supported by the increasingly person-centered design of hearing aid technology. Greater evidence and a more nuanced understanding of the personal need for adoption versus adherence to the use of consumer hearing care devices are required. Research considering consumer hearing aid acceptance behavior rests on behavior modification theories to guide clinical approaches to increasing hearing aid adoption and adherence. However, in the context of complex chronic health management, there may be a gap in how these theories effectively align with the needs of consumers. Similarly, market data indicates evolving consumer behavior patterns have implications for hearing care theory and implementation, particularly in terms of sustained behavior change. This essay proposes that evidence, including theory and application, be strengthened by revising basic theoretical premises of personal experience with complex chronic health, in addition to considering recent changes in commercial contexts.

## 1. Introduction

Hearing loss has various causes and treatment options, which include the sustained use of hearing aids. The leading presentation of hearing loss in adults is age-related sensorineural hearing loss, caused by the degenerative effects of aging on the auditory system. Sensorineural hearing loss impacts the ability of auditory organs from the ear to auditory pathways in the brain (neural) to detect (sense) and process sound effectively [1]. Age-related sensorineural hearing loss is typically a complex chronic condition that impacts the quality of life [2]. Treatment via sustained use of hearing aids involves the use of consumer-centered technology in the form of a hearing aid and is increasingly associated with accompanying user apps. Regardless of the severity of the hearing loss, hearing aids should be worn all day, every day. Wearing hearing aids is supported by a growing array of consumer-centered audiological technology services. Hearing care extends beyond attendance at the audiology clinic. Online consumer-centered and other variations of care services for managing hearing loss are readily available. Today’s support technologies have resulted in a shift from traditional paternalistic clinician-driven models of care to increasingly consumer-centered service delivery. Engaging care models that improve the connection between consumer and clinician is a step towards truly person-centered models of care [3]. The primary symptom of hearing loss is a reduced ability to communicate effectively or with reduced ease. Mild hearing loss, however, may allow a person to feel that their ability to communicate has not been compromised, leading some individuals to choose not to address or treat their condition [4,5,6,7]. Individuals who delay acknowledging their hearing loss may be mistaken, believing they can operate adequately without hearing aids. Regardless of personal perceptions, an individual’s ability to communicate effectively is suboptimal; no matter how well a person may feel they are managing without addressing their hearing loss, there is still a need to treat the hearing loss, usually using hearing aids. As a result, there is a personal conflict between the need for clinical attention and the need for personal behavior change. Even though hearing aid technology and satisfaction with hearing aids have steadily increased over the last 20 years, lack of adherence to recommended interventions remains a significant problem in many complex chronic conditions, and hearing care is no exception [8,9].

Health status is functionally and personally integral to a person’s sense of identity [10,11]. The profound personal impact of complex chronic health conditions has long been understood. Barnard [12] succinctly explained chronic illness as an

“…encounter with limitation and finitude. In particular, chronic incurable conditions require a major adjustment in personal identity. Patients must assimilate the fact of imperfection, of impairment and constraint”[12] (p. 341)

Personal autonomy, agency, and independence are challenged by complex chronic conditions, such as hearing loss [13,14]. Hearing loss can impair a person’s ability to perform tasks. For example, hearing loss affects understanding speech over the phone at a normal volume, responding correctly or promptly to instructions in a workplace or healthcare setting, and socially engaging in dynamic and noisy environments. These impairments have a potential impact on functional well being [15,16]. Living with hearing loss can cause a shift in identity as the sense of self that existed prior to the condition is permanently challenged [17,18].

Illness identity describes how a person perceives their condition and its treatment, as well as how the person integrates their condition into their identity [17,19]. Acceptance or rejection of these changes is associated with improved or avoided self-management, adherence, and self-care behaviors for the condition [19,20,21,22,23]. Experience of complex chronic conditions is one of loss of agency [14]. Imposed changes in functional status and self-identity by having the condition are considered an involuntary experience. Accepting the condition is based on the acknowledgement that there is no choice about having the condition or its presence in one’s everyday life rather than understanding the condition, its causes, or treatment choices [10,24].

Hearing aids are intended to compensate for the diminished ability to hear and communicate. However, it is physiologically impossible to regain 100% functional hearing or a pre-condition capability [25]. There is irreversible physical and potential emotional loss. Hearing loss has involuntary implications on a person’s sense of self and is summarized by the following statement:

“Hearing loss involves change and adjustment and a changing sense of identity that threatens one’s control over one’s autonomy and independence. [Hearing loss] adds to other physiologic changes and to societal attitudes that foster dependency to reduce one’s sense of personal control” [13].

Involuntary changes to personal identity inherently instigate an emotional process of grieving and acceptance [26,27,28,29]. Ageing and retirement, which mark a mature coming-of-age phase of life, are two other major changes that can impact personal identity.

Ease of communication can be improved with hearing aid use, along with reducing the rate of further deterioration of hearing loss and associated impacts on health [16,25,30,31]. For individuals to attain maximum benefit from their prescribed hearing aids, they must be worn on a daily and long-term basis. However, current clinical outcome measures do not commonly address or assess sustained use. Once a hearing aid has been purchased, it is on individuals to actively wear it daily.

Knowledge of the negative consequences of untreated hearing loss is insufficient to motivate sustained behavior change in hearing aid consumers, as there continues to be non-use of hearing aids, even after purchasing them [8,10]. The non-use of hearing aids has contributed to the Consumer Technology Association publishing standards for digital therapeutics, providing evidence of satisfaction and adherence [32]. With the vision of improving consumer adoption and adherence to hearing care, the growing array of technology available is becoming consumer focused [3].

There remains a gap between the advancement of hearing technology to treat and support hearing loss and the increasing number of people who would benefit from wearing hearing aids [33]. Behavior change theories have been used to guide research and practice reasoning to increase the likelihood of seeking help for hearing loss and eventually purchasing hearing aids. Change theories are also being applied to the consumer-centered advancement of product and service development in hearing care [33,34]. The trend toward consumer-centered development of technology and services in hearing care has resulted in a reduction in the delay between the identification of hearing loss and the first-time use of hearing aids [35]. However, the lived experience: the personal traits, beliefs, values, abilities, and aspirations of individuals requiring hearing care and experiencing the loss, may have been ‘deafened’ by society’s preoccupation with marketing campaigns and the acquisition of consumer goods.

The aim of this essay is to synthesize evidence and articulate current knowledge about concepts derived from behavior change theories. To identify and map concepts utilized in practice and reflected in the literature, behavior modification theories often employed in hearing care were identified. The purpose of this essay was not to perform a systematic review or to offer a comprehensive description of all existing approaches but to present evidence to debate whether the hearing care industry should reassess its approaches to hearing aid adoption and long-term hearing care. The provocative question posed in this article is, “Does the inclusion of commonly used behavioral theories in hearing care assist clinician knowledge of the requirements of individual adoption and adherence to hearing aids in today’s environment?”

## 2. Behavior Theories for Treatment of Age-Related Sensorineural Hearing Loss with Sustained Use of Hearing Aids

Analyzing behavior theories provides a benchmark to improve the adoption and adherence of behavior-dependent hearing care treatments and technologies. Traditional hearing care typically comprised seeking a hearing test and advice, which may include referral for further investigation, and purchasing a hearing aid from clinic attendance, guided by one clinician. Innovation in hearing care includes new models of service where hearing aids can be acquired online. However, to support a person using their hearing aids successfully, effectively, and to their greatest potential, it is important to realize that clinical input and guidance are not mutually exclusive to online provision of hearing aids, with such services being further offered, recommended, or required via a variety of innovative business models that uncouple service and device provided for the person to access both in a way that serves them best [36,37,38]. The change in consumer–clinician interaction with online providers can allow removal or avoid repetition of certain help-seeking adoption behaviors: attending a hearing test, inquiring about hearing aids, fitting hearing aids, and, finally, buying them. Traditional adoption behaviors are no longer all required per clinician but instead can be considered per person receiving the services as they pursue their own hearing care journey on their terms.

Long-term habit modification is more than a buying choice. Behavior modification theories for long-term hearing care must account for lived experience to address the elements that drive adherence. Hearing aid users’ ultimate behavior aim is sustained behavior modification. Hearing aid adoption and adherence are essential to ensure consumer ownership and continuing person-centered support for hearing aids to be effective in treating hearing loss.

To understand and discuss opportunities and obstacles regarding hearing aid adoption and adherence processes, behavior change theory applied in audiology research is explored. Table 1 provides an overview of theoretical concepts, describing elements of the application, prerequisite factors, objective descriptors, exclusions, and application focus.

Hearing care literature recognizes six models for behavior change: Capability, Opportunity, Motivation Behavior (COM-B), Trans-Theoretical Model (TTM), Health Belief Model (HBM), Theory of Reasoned Action/Theory of Planned Behavior (TRA/TPB), Common-Sense model (CSM), and Self-Determination Theory (SDT) (See Table 1). COM-B, TTM, and HBM have been the most used models of behavior change for studies in audiology research regarding adoption and adherence over the last 20 years [46,47,48,49]. These seminal behavior change theories were developed from a public health perspective, focusing studies on the design, mapping, and policies of interventions for preventative lifestyle changes (e.g., smoking, exercise, health screening and vaccination) [39,40,41,50]. However, COM-B, TTM, and HBM theories were founded on voluntary lifestyle behaviors, which contrasts with the involuntary nature of the lived experience of hearing loss.

The disparity between the original behavioral change context on voluntary lifestyle behaviors and behavior change theories applied in hearing care research for the complex chronic condition of hearing loss as an involuntary experience raises the question—are these theories appropriate for use with hearing loss? Wearing a hearing aid necessitates deliberately placing one in the ear or on the ear of the person in need, making their usage an active choice. Acting is a skill essential for good hearing aid use. Motivation to act is required in behavioral theories adopted in audiology [46,47,48,49,50,51].

Voluntary behavior and involuntary experience have different contexts due to the contrast between being an active participant versus a recipient of change. The act of wearing a hearing aid is voluntary behavior. However, the lived experience of hearing loss that influences the motivation to keep wearing a hearing aid is involuntary. Behavior change research in audiology repeatedly investigates adoption and rarely adherence to the wearing of hearing aids [27]. This oversight is reflected across discussions of behavior change theories applied in audiology and wider hearing care audiences [47,49,52] (see Table 2). Factors for success in adoption versus adherence differ due to the different contexts of the two processes [53]. Adherence to wearing hearing aids has not been explicitly investigated as a behavior change in hearing care research [53,54].

The emotional sense of loss of function and change of identity described as illness identity, how a person feels about their involuntary complex chronic health change experiences, is under-represented in the use of behavior change theories in hearing care research (see Table 2). The Theory of Reasoned Action (TRA), evolving as the Theory of Planned Behavior (TPB), and the Common-Sense Model (CSM) are three theories that accept emotion and attitude as prerequisite components for the management of any complex chronic condition [42,43,59]. TRA, TPB, and CSM theories are used in managing health behaviors such as medication adherence.

TRA, TPB, and CSM recognize emotion is relevant to desired health behaviors or pertaining to the negative impacts of a condition. However, a person’s emotional relationship with their condition and the impact on such health behaviors is overlooked [42,43,59]. The tension between the original context of the TRA/TPB, HBM, and CSM and their application in hearing care remains. The assumption of rational processes contradicts the emotional nature of involuntary change, loss of function, and self-identity associated with complex chronic conditions such as hearing loss [12,13,14,18].

In CSM, distress caused by a health threat or condition is at the root of an emotional response [44,59]. The fear of, or need for, danger control caused by a health threat is expressed in the shared origins of fear-drive models [60], followed by a parallel processing model [61] that led to the creation of the CSM theory [44]. Fear-based behavior change is supported when individuals exhibit high levels of self-efficacy [62]. However, lack of effectiveness, without complete self-efficacy, has led to fear-based models no longer being supported and considered less appropriate for any context where self-efficacy is challenged [62,63].

Emotional components involved in a fear-driven response relate to the condition’s immediate effect, implying an acute response rather than a chronic perspective on the impact of hearing loss on the person’s self-identity [12,28,29]. This difference can be represented as the difference between phrases “I am anxious that hearing loss will make me miss out on some important information being shared”, placing emotional focus on the condition, as opposed to “I am anxious that I can no longer manage these meetings anymore”, placing the onus on a capable person’s self-identity. This articulates the emotional loss a person with hearing loss experiences [26,27] and aligns with the literature written about adjusting to chronic health conditions, recognizing a loss as an emotional grief-based process [28,29].

The recent revision of the CSM theory acknowledges differences that apply to the management of acute, complex, and chronic conditions [59]. Theoretical concepts of emotional response remain anchored to the condition or treatment, overlooking the impact of the person’s self-identity [17,18]. The discussion of identity representation in theoretical behavior change describes symptoms of the ailment or perceptions of the treatment’s influence on it. In the context of hearing loss, symptoms or perceptions of treatment effects could be demonstrated by an individual adjusting their perception of the severity of the hearing loss or by observing that wearing hearing aids reduces the difficulties in day-to-day communication and can recover a sense of confidence in social environments that contribute to the quality of life [27].

Emotional response to hearing loss is anchored to self-identity rather than the condition or treatment. Individuals recognizing a loss of their hearing, in turn, are faced with a change in self-identity [13,26,27]. The loss being mourned is not necessarily hearing ability but the change in lived experience underpinning self-identity. The distinction between a focus on the condition, or adoption of hearing technology, versus a focus on the person’s involuntary change experience demonstrates the difference between a short-term active decision to buy hearing aids, distinct from the continuing lived experience of hearing loss impacting their daily choice to adhere to wearing hearing aids. Hearing loss involves an acute action and a chronic experience. Adopting or buying a hearing aid is an acute action, whereas adherence to wearing a hearing aid is a chronic process, a change to self-identity.

For the hearing care industry, adopting hearing aids is consumer-focused purchasing behavior. Adherence to the use of hearing aids, on the other hand, is a person-centered behavior. Person-centered care is a popular discussion topic in hearing care, endorsed to improve clinical outcomes and successful behavior change [64,65,66]. However, not until 2022 was a new approach to person-centered outcome design thinking published by Allen et al. (2022) [67]. According to contemporary authors Allen et al., there is a contrast between clinician and consumer perspectives of what is valued by the person’s lived experience of hearing loss [67,68,69]. When listing the most valued outcomes of hearing care, clinicians ranked behavior of self-empowerment as the seventh most important, whereas individuals with the lived experience of hearing loss stated their most valued outcome was the ability to live independently, the ultimate expression of self-efficacy and self-identity [67].

Lack of recognition of self-identity or emotional response to involuntary change caused by health conditions is an observation across hearing care studies into behavior change. The self-determination theory (SDT) applied to hearing loss proposes internal and external processes are required to recognize the impact on motivating people to seek help [56,57]. Internal processes are more strongly predictive than the external process of action taken [56,59]. The Focus of SDT in hearing care remains limited to the adoption of hearing aids rather than sustained adherence.

Behavior changes theories frequently used in hearing care focus on adoption. Behavior change theories in hearing care place the adoption of treatments and technologies such as hearing aids as an outcome [70,71]. This would be fitting for consumer decision-making modeling [72] rather than sustained behavior change required in complex chronic health. Further behavior change theories frequently used in hearing care under-represent the impact of loss, the involuntary change to a person’s self-identity.

The following section discusses recognized behavior change factors in audiology, suggests the potential impact on improving behavior change thinking, and highlights recent changes and evidence in the field of hearing care that may contribute to a better understanding of successful, long-term behavior change in hearing care adherence.

## 3. Discussion

Theories of behavior change date from the 1950s [41,60], before the rise of innovations in hearing care technology and service provision, is seen today. The generational gap between the origin of theory and modern application questions—are these models still the most appropriate to use when discussing contemporary care? The most meaningful change in modern hearing care is an appreciation for self-efficacy, self-management, and balanced control, a productive relationship between the consumer and service provider [3,73].

The concept of self-efficacy was realized in 1977 [74], becoming a cornerstone of change in behavior change theory. In 1983 TTM was developed, building on ideas of self-efficacy [50]; this theory was followed by TRA and revised as TPB [75]. Many models thereafter incorporated recognition of self-efficacy in their foundational design [44,45,76]. COM-B theory is one recent model that recognizes psychological capability as an objective behavioral component rather than the subjective nature of self-efficacy [40]. The examined behavioral change theories suggest efforts have been developed to recognize the impact of self-efficacy on behavior, yet lived experience that influences perception remains overlooked.

Relative to market potential, hearing care continues to demonstrate low rates of adoption, along with persistent issues with adherence to hearing aid use [8,9]. Increasingly consumer-centered technology, service models, and user experience have the potential to help overcome these trends. However, the available evidence is limited and not a focus of research or wider discussion. Consumer-centered technology and health and care management have a mutual goal of improving user experience and health outcomes. Recognition of the consumer as an active participant in their health and care management is the embodiment of person-centered care, promoting the value of empowerment, enabling autonomy, and self-efficacy [77,78,79,80].

Technology developments in hearing care enable a reduced dependence on traveling to clinics to receive services and clinician-controlled health management models and services. Instead, a shift towards empowering consumers to manage their own health and care needs is increasingly possible [3,72,81]. Remote technologies that enable telephone, video, app, and other digitally based activities and interactions are extensively used in complex and chronic health management and demonstrate improved adherence to therapeutic regimes, technologies, and health outcomes [82,83,84,85]. The use of digital enablement with hearing technologies and consumer-centered products and services that empower the hearing care user shows positive effects in improving satisfaction and reducing perceived barriers to managing hearing loss [86,87,88,89].

The importance of consumer satisfaction and perceived self-efficacy is demonstrated by its association with the adoption and adherence to hearing aid use [68,73,90,91]. The increase in person-centered digitally enabled hearing technologies and services in the hearing care industry is recognized as a positive approach for supporting the potential restoration of self-empowerment and self-management, promoting self-efficacy [13,73,92,93]. If the lived experience of complex and chronic conditions is emotionally entrenched in loss, restoring a sense of capability through empowerment, and promoting a restoratively positive shift in self (or illness) identity [17], may be effective in supporting successful and sustained behavior change and adherence [11,13,18,23,94]. Behavior change models in audiology may benefit from incorporating the latest information gleaned from the increasing use of digital enablement and internet-based methods of care to promote self-empowerment and self-efficacy in achieving sustained behavior change and adherence to using hearing aids.

### 3.1. The Impact of Empowerment in Meeting Behavior Change Needs

Online services and the use of self-fit products have been described as empowered hearing care models in hearing care [95,96]. While consumers have access to more empowered hearing care products and service models, self-management or self-efficacy is difficult to measure outside the confines of research settings. However, in today’s empowered hearing care market, satisfaction is a widely available metric that can help analyze consumer behavior and outcomes.

Satisfaction with hearing technology has increased in recent years due to improvements in technology design and performance; yet, there is still much to be achieved or agreed upon regarding improvements in market penetration [35,97,98]. Market surveys indicate adoption rates of first-time buyers of hearing aids have been increasing since 2014 [35,98]. Emergence and growth of products and services that support user empowerment, such as direct-to-consumer hearing technologies and self-fit hearing aids, were established in 2014, evidenced by the first inclusion of non-traditional technology being fitted (non-clinic or clinician dependent) in a field market survey report that year (Blamey Saunders hears circa 2012, see [38]) [99]. The expansion of hearing technologies since 2014 has led to new terminologies to describe hearing devices, for example, Personal Sound Amplification Product (PSAP) or hearable [3,100]. The rationale, unlike hearing aids, hearing devices are not classified as medical devices. Recent market surveys for hearing care have continued to include PSAPs, further supporting the acknowledgement of consumer-centered trends [35,98]. In the United States, until 2022, the Food and Drug Administration Authority (FDA) had restricted hearing aid sales to clinic attendance and clinician-dependent sales channels. Over-the-counter (OTC) purchases of hearing aids have since been authorized under specific conditions for the device to be available for sale for people with mild to moderate hearing loss, whereby a clinic or clinician is not required [101]. PSAPs, on the other hand, are functionally extremely like hearing aids and are available via any consumer sales channel. According to market surveys, by 2019, a trend for purchasing the first hearing device in the form of a PSAP online rather than in-person attendance became more favorable to consumers, going from ‘equally likely’ to ‘nearly twice as likely’ to ‘potentially buy’ [98]. Increased availability of online hearing care supporting resources and technology is a factor in this upward trend. Given hearing aid adoption remains low compared to the size of the available market, it is encouraging that online availability and sales of hearing devices which encourages consumer autonomy are gaining popularity [33,35,98]. A 2022 market survey reported adoption rates for first-time buyers of hearing technology found PSAPs to be significantly higher (83%) compared to hearing aids (31%) [35]. Further, the breakdown of sales channels for hearing aids showed a positive trend between adoption rate and relative user empowerment by product design [35].

Like the theories and audiology applied research discussed so far, the market data available focuses on the adoption of hearing technology rather than adherence. In future, the specialty of hearing care would benefit from adding variables related to adherence to hearing care and technology management in future market surveys and analyses.

### 3.2. Measuring Satisfaction as a Bridge from Adoption to Adherence

Satisfaction is linked to potential adoption and adherence to hearing care [48,95]. Research and market data agree: there is more to capture in understanding what matters in converting the high rates of hearing aid attrition into higher rates of hearing aid adherence [8,9,53]. Consumer-centered digital hearing technologies that support self-management and perceived self-efficacy have been identified as holding potential in this challenging relationship [3,73,81]. The growing range of products and services that can be accessed independently encourages consumer autonomy. PSAPs, self-fit hearing aids, or online providers with or without associated clinics are examples. The longest-running model of hearing care, offering hearing aids online and in a clinic in an interchangeable blended service model, demonstrates comparable product or service satisfaction for online clients [102]. A double-blind clinical trial for hearing aid provision from traditional clinic-based sales or online clinic-independent sales further support comparable satisfaction rates, indicating removal of clinic control at the point of sale is not detrimental to achieving consumer satisfaction [103,104]. Furthermore, rejection or adoption of hearing aids, demonstrated by either choosing to refund the device, extend the trial period, or keep the device beyond the trial period, indicated that autonomy at the point of sale may be beneficial for improving access and adoption of hearing technology with no significant loss in satisfaction [102,103]. Humes et al. included placebo devices, which provided no objective amplification benefit, but satisfaction, adoption, and adherence were still observed, highlighting the powerful impact that autonomy can have in positively influencing hearing care [103,104]. Humes et al. conclude that post the purchase period remains of significant importance in improving continued satisfaction and willingness to adhere to wearing hearing aids [103,104]. Sawyer et al. recommend a need to shift focus away from adoption and toward the emotional state of the person experiencing hearing loss and recovery of self-efficacy [105]. Consumer- and person-centered approaches to behavior change in hearing care should revise current methods toward understanding factors in long-term behavior change considering new evidence and trends [35,37,98,103,104].

Behavioral research, seminal theories, market trends, and the construction of commercially derived research data provide opportunities to revise current thinking and identify opportunities for the implementation of new and improved learnings. Studies in hearing care research highlight the value of adoption-related factors and point to the need to address adherence considerations (see Figure 1). Figure 1 depicts the first component of behavior change as the emotional status of the person in relation to having experienced an involuntary change to their self-identity by the chronic condition. Successful adoption of hearing aids is then impacted by the state of a person’s motivation, attitude, readiness, acceptance, and self-efficacy, along with potentially more factors yet to be identified. After adoption, identified factors implicated for the person to then transition to adherence successfully as an independent stage are satisfaction, self-efficacy, self-identity, self-management, autonomy, and change of attitude. Tools can be provided by the hearing care field to support a person’s journey from the initial emotional status towards sustained adherence, behavior change supporting tools can be provided and utilized at any stage.

Starting a successful hearing care journey is highlighted by revisiting seminal theories of behavior change alongside lived experience of chronic health; the importance of understanding consumer emotional status in accepting a personal change, rather than the condition itself. Commercially derived data that govern technology design, and business models, shaping modern changes to available sales and service models, have the potential to provide real-time data on consumer behavior and outcomes. Combined learning from research and market data indicate that consumer-centered technology is sought by consumers, sales models can enable consumers to access and adopt hearing technologies, and autonomous services models have the potential to positively impact known behavioral factors. In agreement with the Consumer Technology Associations’ recent standards for digital therapeutics, evidence of adherence and satisfaction is considered of value in improving and revising what constitutes success in hearing care [32]. Adherence to therapeutic technologies requires significant research and data to reveal what will matter most. With a greater choice in sales channels, service models, and hearing technology, there is an opportunity to learn directly from consumer behavior-lived experience if the right questions are asked and the right data is collected.

## 4. Conclusions

Clinicians’ understanding of the requirements for adoption and adherence to hearing aid use in today’s hearing care environment are hampered by the limitations of how behavioral theories continue to be applied in hearing care research. Consideration has been given to understanding behavior change theories applied in clinical practice and the consumer market of hearing care. Understanding the personal experience of hearing loss is impacted by an involuntary loss of functionality and change to self-identity has been highlighted. Consideration of the influence of service and product design to support personal needs caused by a loss in a respectful and effective manner is a foundation upon which the new era of digital health intervention models can expand and enable changing consumer behavior in hearing care. Chronic conditions have not changed; however, if we want to see a change in consumer behavior response to them, we should adopt models of care which empower and support those with lived experience of hearing loss as a complex chronic condition.

## 5. Limitations

This essay has provided a stimulating rethinking of issues concerning the successful treatment of age-related sensorineural chronic hearing loss. A systematic review is still needed to further explain how existing techniques for fostering permanent sustained behavior change in persons with hearing loss should be altered or handled. Other forms of hearing loss and their management have not been included in this paper and, thus, should be investigated for additional research, as they may not be relevant to the discussion and recommendations mentioned in this essay. Ongoing research is recommended.

## Figures and Tables

**Figure 1 ijerph-20-05605-f001:**
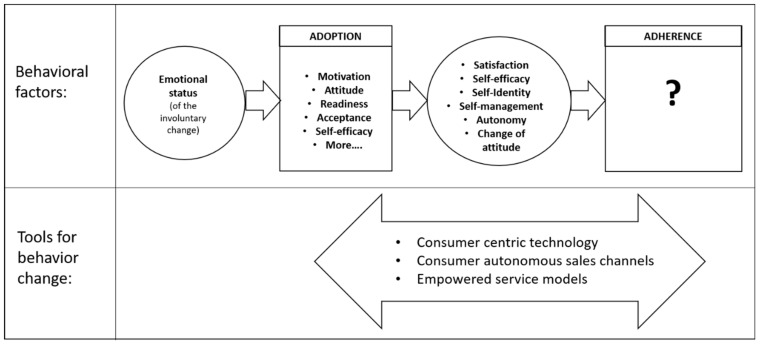
Overview of current knowledge to support sustained behavior change in chronic health management of hearing loss.

**Table 1 ijerph-20-05605-t001:** Behavior change theory applied in audiology research.

Theory	Original Theoretical Application	Pre-Requisites (Factors That Need to Be Present for Any Behavior to Occur)	Objective Description	Exclusion	Application Focus
Capability Opportunity Motivation-Behavior (COM-B)	Lifestyle behavior: Intervention mapping strategy developed in tobacco and obesity studies further developed with climate emergencies, pandemics, violence, and addictive behaviors such as tobacco use (Michie et al., 2018) [39].	Capability, Opportunity, Motivation	Sources of behavioral influence	Emotional status	Voluntary behaviors
Trans-Theoretical Model (TTM)	Lifestyle behavior: Analysis of readiness to quit smoking applied in health promotion programs (Prochaska and Velicer, 1997) [40].	Stages of readiness: precontemplation, contemplation, preparation, action, and maintenance, whereby maintenance is considered a 6-month window and relapse return to an earlier stage in the cycle.	Readiness to adopt intervention/behavior change.		
Health Belief Model (HBM)	Vaccination and screening for TB prevention campaigns (Hochbaum et al., 1952) [41].	Individual beliefs of perceived: susceptibility, severity, benefits, and self-efficacy regarding health status and/or the intervention.	Motivation towards help-seeking behavior/cue to action.		
Theory of Reasoned Action (TRA)	Health behavior; used mostly on sexual behavior, vaccinations, and exercise, i.e., specifically, voluntary behaviors considered lifestyle and disease prevention (Fishbein and Ajzen, 1980) [42].	Attitudes and subjective norms combine with behavioral intentions to predict behavior.	Motivation to act based on expectations.	Emotional response to the prescribed treatment plan.	Adoption of the prescribed treatment plan.
Theory of Planned Behavior (TPB)	Further development if TRA’s description of norms and intentions in line with modern concepts of self-efficacy at the time (Ajzen, 1991) [43].	Addition of perceived control to TRA. Perceived ability to act is considered only with external perceptions.	Attitude/motivation to act.	Emotional response to having the condition.	
Common Sense Model (CSM)	Fear driven focused on the lifestyle behavior of smoking and disease prevention as a threat of tetanus infection (Leventhal et al., 2003) [44].	Cognitive representation of the condition and emotional representation of the fear response to the health threat combine to determine chosen coping strategy, followed by evaluation of the coping response to decide any adaptation or new decision making.	Adoption of coping strategies in response to a health threat.	Individual loss, the involuntary change to self-identity caused by having a complex chronic condition.	Voluntary behavior
Self Determination Theory (SDT)	Motivation and personality pertain to motivation (Ryan and Deci, 2000) [45].	Extrinsic factors and intrinsic factors, whereby 3 key intrinsic motivators, i.e., psychological needs, are needed for self-initiated behavior; Autonomy, (perceived) Competence and Relatedness.	Motivation: Internal processes shaping motivation for adoption.	Emotional response to having the condition.	Adoption of the prescribed treatment plan.

**Table 2 ijerph-20-05605-t002:** Applications of behavior change models in Audiology research.

Theory	Behavioral Outcomes Found	Considerations
Capability Opportunity Motivation-Behavior (COM-B)	Behavior planning for the person was identified as an intervention strategy to encourage the formation of the habit to continue to wear hearing aids (Barker et al., 2016) [54]. Applied to rehabilitative support for hearing aid acclimatization, improved competency skills and knowledge are important in successful hearing aid use (Ferguson et al., 2016b) [46].	The application of the model only extended to the fitting appointment with the remaining strategic components assigned to the clinician in the form information provision (Barker et al., 2016) [54]. COM-B-designed rehabilitative support, while effective for skills development, did not lead to any improvement in hearing aid use (Ferguson et al., 2016b) [46].
Trans-Theoretical Model (TTM)	Attitudes and beliefs that aligned with stages of change, or readiness, showed a correlation with help-seeking behavior (Laplante-Levesque et al., 2013; Saunders et al., 2016a, 2016b) [47,49,52]	The objective acceptance of the health condition, i.e., being able to acknowledge and understand that a person has hearing loss, used to indicate readiness; however, it did not align with adherence to hearing aids.
Health Belief Model (HBM)	Seeking and adopting hearing aids aligned with noted changes in attitude and beliefs following help seeking behavior (Saunders et al., 2016a, 2016b) [49,52]. Greater self-efficacy or beliefs were a strong predictor of help-seeking behavior.	Adherence was not addressed, thus limiting the validity of the interpretations towards sustained behavior change which is currently a significant issue in the field.
Theory of Planned Behavior (TPB)	Seeking a hearing test within 6 months was the only objective measure for which the model did show predictive strength, especially for identifying individuals who may need more intervention planning/action (Arnold et al., 2019) [55].	Attitude change post-fitting was proposed as a factor in the successful transition from adoption towards adherence, suggesting a transitional process may be more appropriate (Arnold et al., 2019) [55].
Self Determination Theory (SDT)	Self-reported hearing difficulty as an intrinsic motivation was associated with hearing aid adoption (Ridgway et al., 2015) [56].	No association between support of autonomy (core psychological need in the model) and hearing aid adoption was found, while adherence was beyond the scope of the studies (Ridgway et al., 2016, 2017) [57,58].
Common Sense Model (CSM)	Emotional distress was able to be aligned into types of coping strategies that impact the person’s experience with their hearing loss because of distress caused by having the loss rather than having the loss itself (Bennett et al., 2021) [26].	An understanding of the lived experience is needed to allow emotional responses to be considered as part of a chronic disease model, with the aim to improve outcomes (Bennett et al., 2021) [26].

## Data Availability

Not applicable.

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
