# Peer review of "Behavior Change in Chronic Health: Reviewing What We Know, What Is Happening, and What Is Next for Hearing Loss"

_ijerph, 2023, doi:10.3390/ijerph20085605_

Round 1
Reviewer 1 Report
The authors present a detailed review on how behavioral theories are applied in dealing with hearing care research. This has important consequences in the way clinician understand the requirements for adoption and adherence
to hearing aid use. The paper is clearly written and it makes a significant point to the issue of hearing loss treatments today. I may suggest its publication in the journal.
I have some minor points listed below, and a rather general question regarding Fig.1. I had trouble in following the different parts discussed in the figure. I would suggest the authors to expand considerably the caption explaining the meaning of each part in simple words and in more detail.
Minor points:
(1) Line 187: one 'are' too much.
(2) Lines 207-243: Maybe you should consider splitting this long paragraph into subparagraphs carrying a different concept or idea. This, in keeping the reader focused on a single concept at a time, facilitating its comprehension.
(3) Lines 244-282: same as in (2).
(4) Check line 274, there is a wrong parenthesis. It would be better if you put together author's name (Allen et al.) with Ref. number [64].
(5) Line 308: same comment as in (1). Please check it.
(6) Lines 324-360: same as in (2) and (3).
Author Response
Please see attachment, thanks.

Reviewer 2 Report
First, I would like to thank you for reading and reviewing this manuscript. The manuscript deals with an important topic, i.e., the behavioural changes in terms of hearing aid users, which is particularly essential considering patients’ long-term compliance. However, my impression is that the presentation of previous outcomes is too general, and sometimes the interpretation of the ideas is somehow ’black and white’. Therefore, I do not find the manuscript sufficient for publication in its current form. Find my specific comments and recommendations below.
Line 13. Here is the first appearance when the authors talk too generally. Hearing loss is a broad diagnostic group, of which the chronic sensorineural hearing loss types can be managed using hearing aids. In the acute cases of sensorineural hearing loss, and conductive types, when middle ear surgery might be necessary, there is no evident indication for using hearing aids. Please try to emphasise this.
Line 34. The same is true here, as mentioned above. Hearing loss is not always a chronic condition; when acute cases are appropriately treated using medicines, they can be completely resolved and will not become chronic. Furthermore, I find it important to state that besides the quality of life, it significantly affects sufferers’ day-to-day functioning. In most cases, quality of life is not assessed in everyday clinical practice because to measure it, the use of quality of life questionnaires is necessary. Management of hearing loss only involves using consumer-centric technologies, e.g., hearing aids in the case of chronic sensorineural hearing loss. Sometimes, when the patient does not want to undergo surgery, of course, we can also offer hearing aids for conductive hearing loss. However, in these cases, certainly, a reduced effectiveness of the treatment is suspected. Therefore, this significantly affects patients’ adherence and adoption; hence, I do not find it appropriate to mix these groups. Overall, I recommend presenting this background information more accurately in the introduction. Furthermore, later, the authors also mention that following hearing loss tinnitus, may also occur. According to previous literature, and the personal practice of the reviewer, patients report tinnitus as more disturbing and a more expressed negative effect regarding day-to-day functioning than in terms of hearing loss. Tinnitus in the case of hearing rehabilitation is crucial, and significantly affects hearing aid-wearing habits. Moreover, I also miss some facts regarding the aetiology of hearing loss, e.g., it is particularly important to state that the incidence of noise-induced and music-induced hearing loss is increasing. WHO has also highlighted the importance of hearing loss in young caused by personal listening devices [Hussain T.; Chou C., Zettner E., Torre P., Hans S., Gauer J., et al. (2018). Early indication of noise-induced hearing loss in young 463 adult users of personal listening devices. Ann Otol Rhinol Laryngol 2018, 127, 703-709]. Consequently, the authors can emphasise why hearing loss is a public health issue.
Lines 46-48. This sentence is too general; hearing loss not necessarily impacts a person’s communication, as it depends on the frequency spectrum involved and the severity of hearing loss. For example, mild hearing loss occurring only in the higher frequencies, subjectively, will not cause any problems regarding everyday communication. Although, it may result in a high frequency tinnitus, but also not necessarily.
Line 71. …routine tasks… Could the authors present some examples? Overall, the quality of the manuscript would be significantly improved with such specific examples. It is easier to understand the intended meaning than reading only general statements for the reader.
Lines 103-104. To provide maximal benefit, other factors are also vital, such as an appropriate hearing test, sampling in the external canal for best fitting, and the adjustment of the hearing aid for the best functioning.
Line 144. Please indicate in the title of this paragraph that here the behaviour therapies regarding hearing loss therapy or management are discussed.
Line 148. In this case, the diagnostic for the background of hearing loss is also crucial. Therefore, other tests, such as a brain MRI, might also be necessary, depending on the clinical presentation of the hearing loss. Therefore, the role of a clinician is more complex than presented here.
Lines 150-151. I find this statement a little bit inappropriate, and I recommend the authors talk more carefully. Buying hearing aids online without medical assistance might be dangerous, and can result in further deterioration of the patient’s hearing. Before giving a hearing aid, a detailed ear, nose, and throat and audiological examinations are necessary, with other examinations in some cases. Without correctly checking the patient’s status, the hearing aid will not effectively work; moreover, it may result in complications. I understand that online hearing aid buying can present positive effects regarding compliance and is important in terms of self-efficacy. Although, the lack of medical assistance might be threatening.
Table 1. Indicate the explanation of the abbreviations in the table/and table caption as well. Therefore, it will be easier to interpret them for the reader.
Lines 236-240. Examples like this would improve the quality of the manuscript and make the reader interested.
Lines 255-256. This is another reason why the authors should include some facts regarding tinnitus in the introduction.
